# Fluctuation Relation for the Dissipative Flux: The Role of Dynamics, Correlations and Heat Baths

**DOI:** 10.3390/e26020156

**Published:** 2024-02-11

**Authors:** Xubin Lin, Lamberto Rondoni, Hong Zhao

**Affiliations:** 1Department of Physics, Xiamen University, Xiamen 361005, China; 2Department of Mathematical Sciences, Politecnico di Torino, 10129 Turin, Italy; 3Istituto Nazionale di Fisica Nucleare (INFN), Sezione di Torino, 10125 Turin, Italy

**Keywords:** fluctuation relation, heat bath, correlation

## Abstract

The fluctuation relation stands as a fundamental result in nonequilibrium statistical physics. Its derivation, particularly in the stationary state, places stringent conditions on the physical systems of interest. On the other hand, numerical analyses usually do not directly reveal any specific connection with such physical properties. This study proposes an investigation of such a connection with the fundamental ingredients of the derivation of the fluctuation relation for the dissipation, which includes the decay of correlations, in the case of heat transport in one-dimensional systems. The role of the heat baths in connection with the system’s inherent properties is then highlighted. A crucial discovery of our research is that different lattice models obeying the steady-state fluctuation relation may do so through fundamentally different mechanisms, characterizing their intrinsic nature. Systems with normal heat conduction, such as the lattice $\phi^{4}$ model, comply with the theorem after surpassing a certain observational time window, irrespective of lattice size. In contrast, systems characterized by anomalous heat conduction, such as Fermi–Pasta–Ulam–Tsingou-β and harmonic oscillator chains, require extended observation periods for theoretical alignment, particularly as the lattice size increases. In these systems, the heat bath’s fluctuations significantly influence the entire lattice, linking the system’s fluctuations with those of the bath. Here, the current autocorrelation function allows us to discern the varying conditions under which different systems satisfy with the fluctuation relation. Our findings significantly expand the understanding of the stationary fluctuation relation and its broader implications in the field of nonequilibrium phenomena.

## 1. Introduction

The 1990s witnessed the advent of fluctuation relations by authors like Evans, Gallavotti, Jarzynski and Crooks, revolutionizing statistical physics and nonequilibrium thermodynamics [1,2,3,4,5,6]. These relations extended the second law of thermodynamics into a statistical–physical framework, embracing both increases and decreases in entropy. Their ratios of positive and negative entropy events are shown to rise exponentially with larger measurement times and system sizes, seamlessly bridging micro and macro thermodynamic theories and earmarking the fluctuation relation as a pivotal development in nonequilibrium statistical physics. Transient fluctuation relations for **time-reversible** dynamics are exact for systems of any size, near and far from equilibrium, and over any observation time [4,5,7,8,9]. On the other hand, **steady-state** fluctuation relations, apparently similar or directly derivable from transient relations, are only asymptotically valid hence not exact, and require proper conditions to be verified concerning the time correlation function of the dissipation [8,10,11]. We will see, however, that exact relations can be verified at any finite time, when the proper correlation functions can be evaluated, cf. [8,12].

Nonequilibrium steady-state heat flow fluctuations are very convenient for testing the fluctuation relations. Under the local equilibrium hypothesis, local entropy generation links directly to heat flow, connecting to the system’s total entropy change. Lepri, Livi and Politi’s 1998 application of fluctuation relations for the dissipation function [13] to nonequilibrium steady-state heat flow fluctuations, validated by numerical simulations in the Fermi–Pasta–Ulam–Tsingou(FPUT)−β lattice model, laid the groundwork for further exploration [14]. Subsequent studies by Van Zon, Cohen, Jarzynski and Dhar et al. expanded this theorem’s applicability, to a variety of systems, including the harmonic oscillator chain [10,15,16,17,18]. These findings, however, raise a key question: does compliance reflect the system’s dynamical characteristics or the influence of the heat bath? Furthermore, the need for larger time windows in larger systems seems to contradict the expectation that larger systems should more readily adhere to thermodynamic laws.

Our research addresses these complexities by examining three distinct one-dimensional lattice models: the harmonic oscillator chain, FPUT-β and the $\phi^{4}$ lattice model, each representing a unique thermal conduction class. The harmonic oscillator chain lacks chaotic motion, exhibiting a system-size proportional thermal conductivity, κ∼*N*. The FPUT-β and $\phi^{4}$ lattice models, with chaotic motion, show divergent and size-independent thermal conductivity, respectively [19,20,21,22,23,24]. This divergence in heat conduction behaviors presents a compelling study contrast.

We demonstrate that, for a fixed system size, each model satisfies the steady-state fluctuation relations for the dissipation function within sufficiently large observational time windows [25], albeit through differing mechanisms. In lattices with anomalous heat conduction, compliance stems from the heat baths at the chain ends, where bath fluctuations permeate the entire chain given adequate time. Conversely, in lattices with normal heat conduction, bath fluctuations diminish rapidly, and theorem compliance is dictated by the system’s statistical physical behavior. Notably, when the observation time is scaled with the characteristic correlation time, a universal, size-independent fluctuation relation emerges. This discovery underscores the correlation’s disappearance as indicative of steady-state fluctuation relations satisfaction, and enabling predictions of fluctuation relations in longer chains based on shorter ones.

The structure of this paper is organized as follows. The next section presents an introduction to the fluctuation relation for dissipation function. We derive the steady-state fluctuation relation from the transient fluctuation relation, building on the foundational work of Evans and Searles [9]. This derivation particularly emphasizes the connection between the fluctuation relation and the decay of correlations caused by dissipation [8]. In Section 3, we introduce the models that will be examined in our study. Section 4 is dedicated to reporting our principal findings. It shows that there is indeed a variety of possible behaviors determining the decay of correlation, revealing the physical mechanisms leading to the satisfy of the steady fluctuation at different levels. Finally, Section 5 provides a comprehensive summary of our work.

## 2. Steady-State Fluctuation Relation

The authors of Ref. [1] considered the Gaussian iso-energetic SLLOD model of a shearing fluid, for *N* particles in *d* dimensions, cf. Appendix A, and proposed and tested the following fluctuation relation:(1)$$\frac{\mu_{i}}{\mu_{i^{*}}}=\frac{\exp\left[-\sum_{n}^{+}\lambda_{i,n}\tau \right]}{\exp\left[-\sum_{n}^{+}\lambda_{i^{*},n}\tau \right]}=\exp\left[Nd\tau {\bar{\alpha }}_{i,\tau }\right]$$
inspired by the theory of Anosov dynamical systems, cf. Appendix B, which they adapted to the time-reversal invariant SLLOD equations of motion. In this formula, $i,i^{*}$ represent conjugate phase space trajectory segments of length τ, $\lambda_{i,n}$ is the *n*-th finite-time Lyapunov exponent computed along the segment *i* in a time τ, and ${\bar{\alpha }}_{i,\tau }\propto -\sum_{n}\lambda_{i,n}$ is proportional to the average energy dissipation rate, along one trajectory segment of duration τ. Then, $\mu_{i}$ is the steady-state probability of trajectory segments of length τ yielding the same ${\bar{\alpha }}_{i,\tau }$, and $\mu_{i^{*}}$ is the probability of segments yielding the opposite value, $-{\bar{\alpha }}_{i,\tau }$. The result, according to the theory of large deviations, is then verified by taking larger and larger averaging times τ. In local thermodynamic equilibrium, Equation (Equation 1) quantifies the second law for reversible dynamics, which is what made it initially very popular. Consequently, in near equilibrium situations it yields linear response with Onsager and Green–Kubo relations [26,27]. Moreover, its derivation from pure dynamics makes the fluctuation relation suitable to go beyond the local thermodynamic equilibrium regime, and of particular interest in the field of small and low-dimensional systems. In that realm, unlike the macroscopic one, fluctuations are observable and comparable to observed average signals. The work on steady-state fluctuations led to the transient fluctuation relation, derived in 1994 by Evans and Searles [9], and that opened the way to a different understanding of the steady-state one, not based on the Anosov assumption [26], cf. Appendix C. Here, we concisely illustrate this approach, which explains our findings.

One fundamental property of the molecular dynamics systems for which the fluctuation relations have been derived is that they are dissipative, which allows systems to reach steady states, but time-reversal invariant. This can be abstractly described as follows. Let M be the phase space and $S^{t}:M\to M$ evolution in M, so that $S^{t}\Gamma$ is the phase Γ reaches at time *t*. In the cases of our interest, $S^{t}$ yields the solution of the equations of motion $\dot{\Gamma }=G\left(\Gamma \right)$. For instance, given the one-dimensional harmonic oscillator equations:(2)$$\dot{q}=p\;,\;\dot{p}=-q$$
one has Γ=(q,p), and:(3)$$S^{t}\Gamma =S^{t}\left(q,p\right)=\left(q\cos t+p\sin t,-q\sin t+p\cos t\right).$$The dynamics are called dissipative if the time average of the divergence of the vector field
(4)Λ=divG
is negative. When this is the case, the system is not Hamiltonian. In particular, negative Λ means that phase space volumes contract. However, time reversibility may still hold. Indeed, it suffices that the equations of motion are invariant under the inversion of time t↦−t, combined with the transformation
(5)$$i\left(q,p\right)=\left(q,-p\right)\;\;\mathrm{so}\mathrm{that}\;\;i^{2}=\mathrm{Identity},$$
where Γ=(q,p) is the collection of positions and momenta of the particles of the system of interest, cf. Appendix B. This also implies that the trajectories stemming from the initial conditions Γ and $\tilde{\Gamma }=iS^{t}\Gamma$ trace back each other in configuration space for a time *t*, and lead to the following general definition. Then, the dynamics are called time-reversal invariants if there exists one transformation i:M→M whose double application is the identity, such that
(6)$$iS^{t}\Gamma =S^{-t}i\Gamma \;\mathrm{for}\mathrm{all}\;\;\Gamma \in M.$$Such a general definition is motivated by the fact that $S^{-t}\Gamma =iS^{t}i\Gamma$, i.e., that the reversal operation generates the backward evolution, and it was implicit in Ref. [1], because the SLLOD equations are not invariant under the time-reversal operation defined by Equation (Equation 5), but they are under, e.g., the following mapping:(7)$$i\left(x,y,z,p_{x},p_{y},p_{z}\right)=\left(x,-y,z,-p_{x},p_{y},-p_{z}\right).$$

Let $f^{\left(0\right)}$ be an initial probability distribution on M, and define the *dissipation function* as:(8)$$\Omega^{\left(0\right)}=-G\cdot {𝜕}_{\Gamma }\ln f^{\left(0\right)}-\Lambda .$$Its name is justified by the fact that $\Omega^{\left(0\right)}$ represents indeed the dissipated power, if $f^{\left(0\right)}$ is the equilibrium (non-driven) distribution that obeys the same constraints of the driven dynamics [8]. That is the case for the situation investigated in Ref. [1], which is shown in Appendix A. Finally, we denote by
(9)$$O_{t,t+\tau }\left(\Gamma \right)=\int_{t}^{t+\tau }ds\;O\left(S^{s}\Gamma \right)$$
the time integral between the times *t* and t+τ of any observable O, computed along the trajectory starting at Γ. Consequently, we can write:(10)$$\Omega_{t,t+\tau }^{\left(0\right)}\left(\Gamma \right)=\int_{t}^{t+\tau }\Omega^{\left(0\right)}\left(S^{s}\Gamma \right)ds=\ln\frac{f^{\left(0\right)}\left(S^{t}\Gamma \right)}{f^{\left(0\right)}\left(S^{t+\tau }\Gamma \right)}-\int_{t}^{t+\tau }\Lambda \left(S^{s}\Gamma \right)ds.$$The quantity $O_{t,t+\tau }\left(\Gamma \right)$ divided by τ gives the corresponding time average ${\bar{O}}_{t,t+\tau }\left(\Gamma \right)$. With such definitions, the derivation of the transient fluctuation relation is just a few lines.

First, one notices that the corresponding Liouville equation in Eulerian form is expressed by:(11)$${𝜕}_{t}f\left(\Gamma \right)=-{𝜕}_{\Gamma }\cdot \left(f(\Gamma )G(\Gamma )\right)\equiv \Omega \left(\Gamma \right)f\left(\Gamma \right)$$
and in Eulerian form by:(12)$$\frac{df}{dt}=-\Lambda f.$$Therefore, given the initial distribution $f^{\left(0\right)}$, which in general is not invariant for the driven dynamics, one obtains [28]:(13)$$f^{\left(t\right)}\left(\Gamma \right)=\exp\left\{-\Lambda_{-t,0}\left(\Gamma \right)\right\}f^{\left(0\right)}\left(S^{-t}\Gamma \right)=\exp\left\{\Omega_{-t,0}^{\left(0\right)}\left(\Gamma \right)\right\}f^{\left(0\right)}\left(\Gamma \right)$$
for the distribution at time *t*. Then, one introduces the intervals of values around *A* and −A,
(14)$$B_{A,\delta }==\left(A-\delta ,A+\delta \right)\;\mathrm{and}\;\;\;B_{-A,\delta }=\left(-A-\delta ,-A+\delta \right)$$
with small δ>0, and observes that:(15)$$\left\{\Gamma :{\bar{\Omega }}_{0,\tau }^{\left(0\right)}\left(\Gamma \right)\in B_{-A,\delta }\right\}=iS^{\tau }\left\{\Gamma :{\bar{\Omega }}_{0,\tau }^{\left(0\right)}\left(\Gamma \right)\in B_{A,\delta }\right\},$$
which is to say that all initial conditions $\tilde{\Gamma }$ that yield the time average ${\bar{\Omega }}_{0,\tau }^{\left(0\right)}\left(\tilde{\Gamma }\right)=-A$ are obtained from those Γ that yield ${\bar{\Omega }}_{0,\tau }^{\left(0\right)}\left(\Gamma \right)=A$, evolving them for a time τ and applying the reversal operation *i*. Denote by $\mu^{\left(0\right)}\left({\bar{\Omega }}_{0,\tau }^{\left(0\right)}\in B_{A,\delta }\right)$ the probability computed with respect to the initial distribution $f^{\left(0\right)}$ that the average of $\Omega^{\left(0\right)}$ takes values near *A*. Then, the ratio of initial probabilities to find values *A* and −A is expressed by:(16)$$\frac{\mu^{\left(0\right)}\left({\bar{\Omega }}_{0,\tau }^{\left(0\right)}\in B_{A,\delta }\right)}{\mu^{\left(0\right)}\left({\bar{\Omega }}_{0,\tau }^{\left(0\right)}\in B_{-A,\delta }\right)}=\frac{\int_{B_{A,\delta }}f^{\left(0\right)}\left(\Gamma \right)d\Gamma }{\int_{B_{-A,\delta }}f^{\left(0\right)}\left(\Gamma \right)d\Gamma }$$
and introduce the coordinate transformation $\Gamma =iS^{\tau }X$, with Jacobian
(17)$$J_{0,\tau }\left(X\right)=\left|\frac{d\Gamma }{dX}\right|=\exp\left\{\Lambda_{0,\tau }\left(X\right)\right\},$$
which leads to:(18)$$\begin{array}{c} \;\int_{B_{-A,\delta }}f^{\left(0\right)}\left(\Gamma \right)d\Gamma =\int_{B_{A,\delta }}f^{\left(0\right)}\left(iS^{\tau }X\right)\;e^{\Lambda_{0,\tau }\left(X\right)}dX=\\ \;\int_{B_{A,\delta }}f^{\left(0\right)}\left(X\right)\;e^{-\Omega_{0,\tau }^{\left(0\right)}\left(X\right)}dX=e^{-[A+\epsilon (A,\delta ,\tau )]\tau }\int_{B_{A,\delta }}f^{\left(0\right)}\left(X\right)dX \end{array}$$
if $f^{\left(0\right)}$ is even under the application of *i*, i.e., $f^{\left(0\right)}\left(i\Gamma \right)=f^{\left(0\right)}\left(\Gamma \right)$, as appropriate for an equilibrium distribution. The transient fluctuation relation immediately follows:(19)$$\frac{\mu^{\left(0\right)}\left({\bar{\Omega }}_{0,\tau }^{\left(0\right)}\in B_{A,\delta }\right)}{\mu^{\left(0\right)}\left({\bar{\Omega }}_{0,\tau }^{\left(0\right)}\in B_{-A,\delta }\right)}=\exp\left\{\tau \left[A+\epsilon (A,\delta ,\tau )\right]\right\}\;,\;\mathrm{where}\;\;\epsilon \left(A,\delta ,\tau \right)\leq \delta$$
with an error ϵ that is not larger than δ, although it depends on all parameters of the theory. Therefore, ϵ can be made as small as one wants, by taking δ as correspondingly small. This also means:(20)$$\frac{\mu^{\left(0\right)}\left({\bar{\Omega }}_{0,\tau }^{\left(0\right)}\in B_{A,\delta }\right)}{\mu^{\left(0\right)}\left({\bar{\Omega }}_{0,\tau }^{\left(0\right)}\in B_{-A,\delta }\right)}={\langle \exp\left(-\tau \;{\bar{\Omega }}^{\left(0\right)}\right)\rangle }_{{\bar{\Omega }}^{\left(0\right)}\in B_{A,\delta }}^{-1}$$
where the right-hand side means the inverse of the average of the exponential of $\Omega^{\left(0\right)}$ integrated from time 0 to time τ, over the trajectories that yield a value close to *A*, with a difference not larger than δ. Indeed,
(21)$$\frac{\int_{B_{-A,\delta }}f^{\left(0\right)}\left(\Gamma \right)d\Gamma }{\int_{B_{A,\delta }}f^{\left(0\right)}\left(\Gamma \right)d\Gamma }=\frac{\int_{B_{A,\delta }}f^{\left(0\right)}\left(X\right)\;e^{-\Omega_{0,\tau }^{\left(0\right)}\left(X\right)}dX}{\int_{B_{A,\delta }}f^{\left(0\right)}\left(\Gamma \right)d\Gamma }$$
is the conditional average of $\exp(-\Omega_{0,\tau }^{\left(0\right)})$ over the trajectories whose initial conditions lie in $B_{-A,\delta }$, which is to say the average with respect to the distribution
(22)$$f_{B_{-A,\delta }}^{\left(0\right)}\left(\Gamma \right)=\left\{\begin{array}{ccc} \frac{1}{\int_{B_{A,\delta }}f^{\left(0\right)}\left(\Gamma \right)d\Gamma }\;f^{\left(0\right)}\left(\Gamma \right) & \mathrm{if} & \Gamma \in B_{A,\delta }\\ 0 & \mathrm{if} & \Gamma \notin B_{A,\delta }, \end{array}\right.$$
which assigns zero probability to the sets outside $B_{-A,\delta }$.

The transient fluctuation relation is very robust because it is based on minimal assumptions: time reversibility and conservation of probability in phase space. It is an identity that holds exact for all τ>0, for an ensemble of experiments all starting in the initial state characterized by $f^{\left(0\right)}$, analogously to other fluctuations such as the Jarzynski equality.

To derive the steady-state fluctuation relation, we can now start from Equation (Equation 19). First, we move from the probability $\mu^{\left(0\right)}$ to the statistic resulting at time *t*, under the assumption that probability is conserved and moves around the phase space together with the phases $S^{t}\Gamma$. This means that the probability of a set E⊂M at time *t*, $\mu^{\left(t\right)}\left(E\right)$ say, equals the probability at time 0 of the set of points that reach *E* at time *t*, $S^{-t}E$, which can be written as
(23)$$\mu^{\left(t\right)}\left(E\right)=\mu^{\left(0\right)}\left(S^{-t}E\right).$$We will then try to extrapolate the result to the steady state, by letting *t* grow without bounds. With minimal algebra, Equation (Equation 21) yields another exact relation:(24)$$\begin{array}{c} \;\frac{1}{\tau }\ln\frac{\mu^{\left(t\right)}\left({\bar{\Omega }}_{0,\tau }^{\left(0\right)}\in B_{A,\delta }\right)}{\mu^{\left(t\right)}\left({\bar{\Omega }}_{0,\tau }^{\left(0\right)}\in B_{-A,\delta }\right)}=-\frac{1}{\tau }\ln{\langle e^{-\Omega_{0,t}^{\left(0\right)}}\cdot e^{-\Omega_{t,t+\tau }^{\left(0\right)}}\cdot e^{-\Omega_{t+\tau ,2t+\tau }^{\left(0\right)}}\rangle }_{{\bar{\Omega }}_{t,t+\tau }^{\left(0\right)}\in B_{A,\delta }}^{\left(0\right)}\\ \;=A+\epsilon \left(\delta ,t,A,\tau \right)-\frac{1}{\tau }\ln{\langle e^{-\Omega_{0,t}^{\left(0\right)}}\cdot e^{-\Omega_{t+\tau ,2t+\tau }^{\left(0\right)}}\rangle }_{{\bar{\Omega }}_{t,t+\tau }^{\left(0\right)}\in B_{A,\delta }}^{\left(0\right)}. \end{array}$$

Here, the second line yields the expression of the fluctuation relation, in the t→∞ limit, if such a limit exists, and the conditional average behaves properly when τ becomes large. Therefore, although it is derived from an exact transient relation that practically always holds, the steady-state fluctuation relation does not always makes sense, and when it does it does not necessarily correspond to the exponential unbalance of probabilities expressed by the original relation (Equation 1). Indeed, the t→∞ of the conditional average may diverge, or it may yield a non-negligible contribution, even in the τ→∞ limit. When this term constitutes a correction that vanishes as O(1/τ), we have a situation analogous to the one discussed within the Anosov formalism. In all circumstances, it is a relation that only becomes exact when τ grows, unlike the transient fluctuation relation, which is exact for all τ. Another difference between the transient and the steady-state relations is that the second applies also to the fluctuations along a single trajectory, and does not need an ensemble.

**Remark** **1.**
*The last term of Equation (Equation 24) must vanish when the limit τ→∞ is taken after the limit t→∞. This is not only a sufficient condition but it is also necessary, because Equation (Equation 24) is an exact relation. This says that correlations of the dissipation function with respect to the initial distribution follow a given rule, when reversible dynamical systems verify the steady-state fluctuation relation. This behavior of the correlations of the dissipation function constitutes the mechanism implying the validity of the steady-state fluctuation relation. The relation is violated under different conditions [8].*


To understand the significance of the last term in Equation (Equation 24), consider the case in which the correlations with respect to the initial distribution $\mu^{\left(0\right)}$ decay instantaneously. Considering that the conditional average then turns equal to the full average, and that the transient fluctuation relation, with some manipulation, also yields [8]:(25)$${\langle e^{-\Omega_{0,t}^{\left(0\right)}}\rangle }^{\left(0\right)}=1$$
one obtains:$$\begin{array}{ccc} & & \;{\langle e^{-\Omega_{0,t}^{\left(0\right)}}\cdot e^{-\Omega_{t+\tau ,2t+\tau }^{\left(0\right)}}\rangle }_{{\bar{\Omega }}_{t,t+\tau }^{\left(0\right)}\in B_{A,\delta }}^{\left(0\right)}={\langle e^{-\Omega_{0,t}^{\left(0\right)}}\cdot e^{-\Omega_{t+\tau ,2t+\tau }^{\left(0\right)}}\rangle }^{\left(0\right)}\\ & & \;={\langle e^{-\Omega_{0,t}^{\left(0\right)}}\cdot \left(e^{-\Omega_{0,t}^{\left(0\right)}}\circ S^{t+\tau }\right)\rangle }^{\left(0\right)}={\langle e^{-\Omega_{0,t}^{\left(0\right)}}\rangle }^{\left(0\right)}{\langle e^{-\Omega_{0,t}^{\left(0\right)}}\rangle }^{\left(t+\tau \right)}=1, \end{array}$$
where ${\langle \rangle }^{\left(t+\tau \right)}$ represents the full phase space average with respect to $\mu^{\left(t+\tau \right)}$, cf. [8]. This implies that the steady state is verified at an arbitrarily short τ. If the decay of correlations is not immediate with a growing τ, the final result will depend on how they depend on τ. An investigation of these issues was performed in Ref. [29] for color diffusion molecular dynamics. In that case, it was found that the correlations decay, hence the steady-state fluctuation relation is verified. Below, we will see that there is indeed a variety of possible behaviors, determined by the physical mechanisms at work.

Ref. [25] considered the classical many-body system that is in contact with two thermal reservoirs maintained at different temperatures, and proposed the equivalence between thermodynamic entropy and the dissipation equation:(26)$$k_{B}\Omega =\Sigma_{\mathrm{therm}}=\left(\frac{j_{R}}{T_{R}}-\frac{j_{L}}{T_{L}}\right)+\frac{j_{L}-j_{R}}{T_{0}}+O\left(\frac{d^{3}}{dx^{3}}\right),$$
where $j_{R}$ and $j_{L}$ are the instantaneous heat flow between systems and the heat baths, respectively. Here, $T_{0}=1/2\left(T_{R}+T_{L}\right)$.

Based on this equivalence relationship, the above steady-state fluctuation relation for the dissipation function can be extended to the steady-state fluctuation relation of heat flow. At the same time, by integrating the local entropy generation, we can obtain the total entropy generation of the system when the time interval is t as follows:(27)$$k_{B}\Omega_{t,t+\tau }^{\left(0\right)}=\int_{t}^{t+\tau }\Sigma_{\mathrm{therm}}dt\approx J\left(\frac{1}{T_{R}}-\frac{1}{T_{L}}\right).$$When the time interval for observation is τ, the cumulative heat flow of the system is $J=\int_{\tau }jdt$. When the system is in a steady state, the local heat flow is on time average equal everywhere, so $\langle J_{R}\rangle =\langle J_{L}\rangle =\langle J\rangle$. From this, we obtain the steady-state fluctuation relation of heat flow [13,30,31]:(28)$$\lim_{\tau \to \infty }\ln\frac{P_{\tau }\left(J\right)}{P_{\tau }\left(-J\right)}=J\left(\frac{1}{T_{R}}-\frac{1}{T_{L}}\right),$$
where $P_{\tau }\left(J\right)$ is the probability that the cumulative heat flow with the time interval τ takes values near *J*.

Defining
(29)$$F\left(J\right)=\frac{\lim_{\tau \to \infty }\ln\frac{P_{\tau }\left(J\right)}{P_{t}\left(-J\right)}}{\left(\frac{1}{T_{R}}-\frac{1}{T_{L}}\right)},$$
our target is to verify whether the fluctuations satisfy
(30)F(J)=J.

## 3. Models

We study one-dimensional lattice models described by Hamiltonian,
(31)$$H=\sum [\frac{p_{i}^{2}}{2m}+V\left(q_{i}-q_{i-1}\right)+U\left(q_{i}\right)],$$
with *N* particles, where $q_{i}$ and $p_{i}$ are the displacement and momentum of the *i*th particle from the equilibrium position, respectively. For simplicity, the mass of particles take unit value. The terms $V(q_{i}-q_{i-1})$ and $U(q_{i})$ are the nearest neighbor interaction potential and on-site potential, respectively. Fixed boundary conditions are applied to the first and last particles, which are connected by two Langevin heat reservoirs with different temperatures, as illustrated in Figure 1.

We investigate the fluctuation behavior of heat flow for three representative lattice models, namely harmonic oscillator model,
(32)$$V\left(x\right)=\frac{1}{2}x^{2};U\left(x\right)=0,$$
the FPUT-β model,
(33)$$V\left(x\right)=\frac{1}{2}x^{2}+\frac{1}{4}x^{4};U\left(x\right)=0,$$
and the $\phi^{4}$ lattice model,
(34)$$V\left(x\right)=\frac{1}{2}x^{2};U\left(x\right)=\frac{1}{4}x^{4}.$$

The local heat flow at the *i*th particle is defined as [30,31]:(35)$$j_{i}\left(t\right)=\frac{1}{2}ap_{i}\left(f_{i}+f_{i+1}\right),$$
where $f_{i}=-\left(V^{\prime }\left(q_{i}-q_{i-1}\right)+U^{\prime }\left(q_{i}\right)\right)$ represents the force exerted on the *i*th particle. The lattice space, *a*, takes the unit in our simulation. We average the local heat flow of *N* particles to obtain the global heat flow, $j_{g}\left(t\right)$, as
(36)$$j_{g}\left(t\right)=\frac{1}{N}\sum_{i}j_{i}\left(t\right).$$

## 4. Results

In Figure 2, we present the distribution of heat-flow fluctuation for the three models with N=128 and τ=100. Throughout our simulation in this paper, we fix the average temperature, *T*, of two heat baths as 0.5 and the temperature difference, ΔT, as 0.1. We note that they exhibit Gaussian-like distribution for all three models, with no significant difference. Gaussian-like distributions of heat-flow fluctuation have been observed in [13], and their reason has been discussed theoretically [32,33].

Figure 3 shows F(J) vs. *J* as a function of τ for several lattice sizes for the three models. It can be observed that, at short time intervals, the prediction of the steady-state fluctuation relation is violated, although F(J) and *J* shows a linear dependence. When τ is large enough, the prediction is always approached for all three models.

However, there are significant disparities. In the case of the harmonic oscillator model and the FPUT-β model, we require increasingly larger values of τ to align with the theoretical predictions as the chain’s length grows. Notably, for a given length, the time window required is greater for the former than for the latter, as illustrated in the first two lines of Figure 3. This is in line with the integrability of the purely harmonic chains, which are at the opposite end of thermodynamics. It is important to highlight that testing the theorem becomes quite challenging for lengthy chains, given the infrequent occurrence of negative heat flow events. Consequently, one must allow the system to evolve for an extended period to obtain an effective distribution function. Indeed, when dealing with the harmonic model, this task is nearly beyond the computational capacity for N>1000. As a result, direct observation of results converging to the theoretical prediction becomes infeasible. From a theoretical standpoint, even though these models can eventually fit the prediction, our analysis reveals that they defeat conventional wisdom. According to this, which larger systems adhere better to the standard statistical descriptions characterizing thermodynamic systems? This, perhaps perplexing, point will be clarified by the analysis of the other models. What is more, we observe a counterintuitive trend in this context, which may be related to the growth fluctuation magnitude with *N*.

For the $\phi^{4}$ model, once the size of the chain exceeds a threshold, τ=200 in this case, the dependence of F(J) to *J* for a given τ becomes identical, as shown in the last line of Figure 3. As a result, for this model, the steady-state fluctuation relation can be generally satisfied for relatively long chains (with about N>200) and relatively large τ (with about τ>200). In practical applications involving real-world materials and measurable time scales, one can conclude that the $\phi^{4}$ model consistently satisfies the theorem.

To understand the mechanism, we calculated the correlation function $C\left(t\right)=\left(\langle J(t)J(0)\rangle -{\langle J(0)\rangle }^{2}\right)/\left(\langle J{\left(0\right)}^{2}\rangle -{\langle J(0)\rangle }^{2}\right)$ of heat flow fluctuation of the three models under a nonequilibrium steady state. We observe that, although C(t) decays to zero with the increase of time, the behavior is different for each model. In the case of the harmonic model, C(t) decays linearly with time, and the rate of decay decreases as a function of the chain’s length. In the FPUT-β model, the function of decay is no longer linear, but the rate of decay still decreases with the increase of the chain length. In the $\phi^{4}$ model, however, C(t) decays very rapidly compared with the other two models. Particularly, it becomes size-independent when the chain length exceeds a threshold, N∼256. We define $t_{r}=\int C\left(t\right)dt$ to measure the cumulative correlation. Figure 4d shows $t_{r}$ as a function of *N*. It can be seen that, when the system size is larger, $t_{r}$ scales as $N^{\gamma }$, with γ values of 1, 0.5 and 0 for the three models, respectively. Interestingly, these scaling relations match those observed in their thermal conductivity dependence on chain size, where $\kappa ∼N^{1}$, $∼N^{0.5}$ and $∼N^{0}$ for the three models, respectively.

We argue that the behavior of C(t) is a result of the interplay between the heat baths and the system’s dynamics. In the harmonic model, we know that $C^{\prime }\left(t\right)=1$ in the absence of a heat bath, since it is an integrable model. In the present of heat baths, the fluctuation from the heat baths propagates to the system at the speed of sound, which is a unit with dimensionless parameters in this context. As these fluctuations are inherently random, they are expected to induce the decrease of C(t). For a rough estimation, we suppose that, once the fluctuations are transported, the correlation of that part of the chain disappears, and the correlation vanishes after t=N. Therefore, $C\left(t\right)∼C^{\prime }\left(t\right)\left(1-t/N\right)$. Here, (1−t/N) characterize the decrease of the region of the sum in the equation. Here, we suppose that the fluctuation spread out with the sound velocity, and thus t=N. As a consequence, $t_{r}∼N$. This estimation aligns well with Figure 4a,d. The value of $t_{r}$ defines the characteristic time that C(t) vanishes, since C(0) is normalized to unity. Therefore, once the fluctuations spread throughout the system over a timescale of $t_{r}∼N$, the correlation vanishes, and the fluctuation relation applies in accordance with the theory of the dissipation function [8,29,34,35].

In the FPUT-β model, it is widely recognized that the system exhibits chaotic motion, which can induce the decay of the correlation function. Nevertheless, the decay follows a power-law behavior, with $C^{\prime }\left(t\right)∼t^{-0.5}$, as evidenced by studies in the equilibrium state [30,36,37,38]. Therefore, the behavior of C(t) results from both the effects of the heat bath and the system itself, which can be estimated as $C\left(t\right)∼C^{\prime }\left(t\right)\left(1-t/N\right)$. This estimation yields $t_{r}∼N^{0.5}$. These results are consistent with Figure 4b,d. Due to the non-vanishing correlation within the system, the heat baths continue to play a role in eliminating the correlation, resulting in a timescale of $t_{r}∼N^{0.5}$ to ensure the system satisfies the fluctuation relation.

In the $\phi^{4}$ model, the system is also chaotic but the fluctuation of heat flow decays as $C^{\prime }\left(t\right)∼e^{-t}$. Even without attributing to the heat baths, C(t) decays to zero before t=N. This explains the results of Figure 4c. Consequently, once the chain length exceeds a threshold, the correlation vanishes after a critical time that is independent of the system size, and, thus, the fluctuation relation applies hereafter.

The reason why the fluctuation relation is satisfied after the vanishing of the correlation heat flow fluctuations is as follows. When there is no correlation between the fluctuations, the distribution of fluctuations appears as a Gaussian distribution $P_{t}\left(J\right)=ce(-\frac{{\left(J-\langle j\rangle t\right)}^{2}}{2\sigma^{2}})$, according to the central limit theorem [32,33,39]. We then obtain $\ln\frac{P_{t}\left(J\right)}{P_{t}\left(-J\right)}=\frac{{\left(J+\langle j\rangle t\right)}^{2}-{\left(J-\langle j\rangle t\right)}^{2}}{2\sigma^{2}}=\frac{2\langle j\rangle tJ}{\sigma^{2}}$. With $\sigma^{2}=\frac{2\langle j\rangle }{\left(\frac{1}{T_{R}}-\frac{1}{T_{L}}\right)}t$, we obtain $\ln\frac{P_{t}\left(J\right)}{P_{t}\left(-J\right)}=J\left(\frac{1}{T_{R}}-\frac{1}{T_{L}}\right)$, i.e., the steady-state fluctuation relation.

Based on the foregoing analysis, we can re-scale F(J) to render it system-size independent. Denote the slope of the F(J) curve in Figure 5 as k(N,τ). We re-scale τ with respect to $t_{r}$ and plot $k(N,\tau /t_{r})$ as a function of $\tau /t_{r}$ in Figure 5. It can be observed that $k(N,\tau /t_{r})$ manifests as a system-size independent curve, $k\left(t\right)=c_{1}t^{-a}+e^{b/t}$. As such, it can be regarded as an extension of the fluctuation relation for an arbitrary τ. The limit of $k(N,\tau /t_{r})$ approaching unity as τ tends to infinity indicates the satisfaction of the original steady-state fluctuation relation. The universality of the harmonic oscillator model is exemplary, and the other two are also fundamentally consistent for larger sizes. Technically, one can employ the universal curve to extrapolate the results obtained for a fixed chain length to other sizes without the need for further calculations when the scalings apply.

The relationship of $k(N,\tau /t_{r})$ with τ arises from persistent correlations. Notably, the global heat flow distribution retains its Gaussian character, even at smaller values of τ. This phenomenon is explained by the process of calculating global heat flow: the averaging of local heat flows from *N* particles, each exhibiting random heat flow characteristics. Consequently, the global heat flow represents an aggregation of numerous small, random variables, aligning with a Gaussian distribution, as dictated by the central limit theorem. The variance of the instantaneous heat flow distribution, measured over a unit time, is denoted as $\sigma_{0}^{2}$. Taking the harmonic chain as an example, and assuming a steady heat flow correlation over an exceedingly brief τ (C(t)≈constant, while neglecting decay due to heat baths), we find that the variance in heat flow fluctuations is represented by $\sigma {\left(\tau \right)}^{2}=\tau^{2}\cdot \sigma_{0}^{2}$. Hence, k(N,τ) approximates $\frac{4\langle j\rangle \tau }{2\sigma {\left(\tau \right)}^{2}}\propto \tau^{-1}$, in agreement with the power law index (−0.96) observed for a small τ in Figure 5a. The τ-dependence of $k(N,\tau /t_{r})$ in the other two models similarly stems from the enduring correlations of heat flow fluctuations.

## 5. Conclusions

The realization of the steady-state fluctuation relation is attributed to the elimination of heat flow fluctuation correlations. In systems characterized by anomalous heat conduction, such as the harmonic oscillator chain and the FPUT-β chain, these heat fluctuation correlations exhibit persistence over extended durations. Thus, the theorem’s realization is critically dependent on the influence of heat baths, which play a key role in dissipating the system chain’s correlations. Although the heat baths’ fluctuations are initially uncorrelated, they facilitate the eventual vanishing of correlations within the system chain as they propagate through it. Consequently, extending the observation interval, τ, becomes necessary to validate the theorem’s applicability, especially when lengthening the chain with heat baths at each end. In systems exhibiting normal heat conduction, like the $\phi^{4}$ lattice chain, the intrinsic heat flow correlation of the system reduces exponentially to zero, thus fulfilling the fluctuation relation. Hence, in chains surpassing a critical length, the theorem’s fulfillment is observable shortly after a minimal transitional phase.

Strictly speaking, satisfaction of the steady-state fluctuation relation is feasible only in systems demonstrated to possess normal statistical properties. The theorem’s widespread fulfillment is largely driven by the heat bath effect. This study thus provides an important reminder on how to properly test the fluctuation relation.

## Figures and Tables

**Figure 1 entropy-26-00156-f001:**
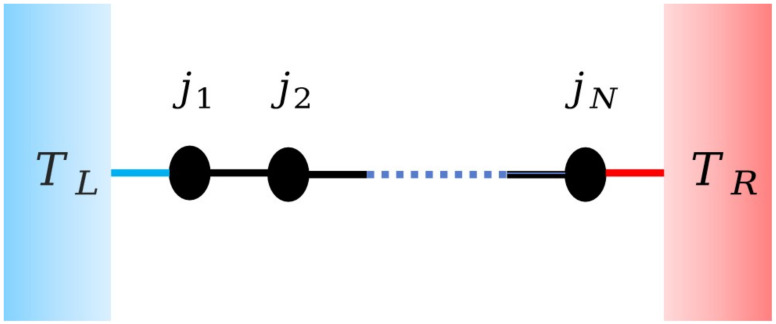
One-dimensional lattice model of Langevin heat reservoirs with different temperatures connected at both ends.

**Figure 2 entropy-26-00156-f002:**
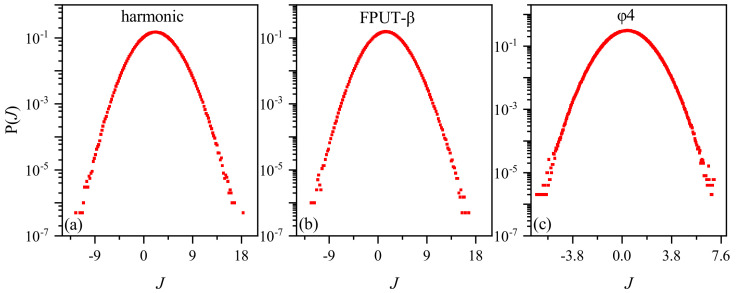
Heat flow distribution across three models. System parameters: consistent across all models. The average temperature at both ends of the heat reservoir T=0.5, and the temperature difference ΔT=0.1. The trajectory segment of length τ is 100. (**a**) The harmonic oscillator model. (**b**) The FPUT-β model. (**c**) The $\phi^{4}$ model.

**Figure 3 entropy-26-00156-f003:**
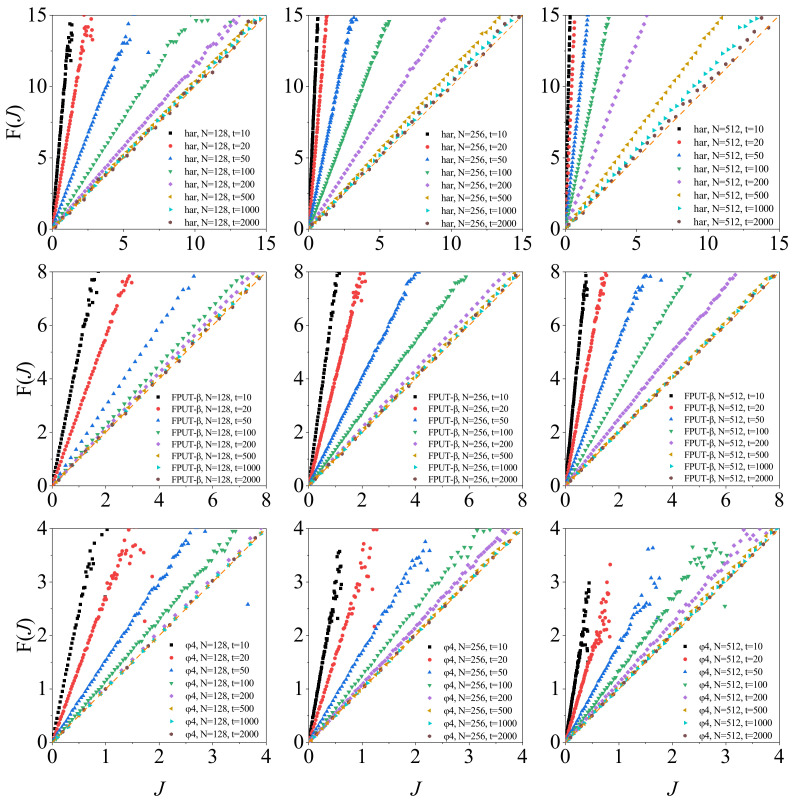
Global flow fluctuation analysis over increasing time intervals in diverse models. System parameters: consistent across all models. The average temperature at both ends of the heat reservoir T=0.5, and the temperature difference ΔT=0.1. The time intervals and particle numbers are annotated in the plots. The first row depicts the F(J) curves for harmonic oscillator models. The second row depicts the F(J) curves for FPUT-β models. The last row depicts the F(J) curves for $\phi^{4}$ models. The results of molecular dynamics simulation are solid points, and the curve predicted by the fluctuation relation is an orange dashed line.

**Figure 4 entropy-26-00156-f004:**
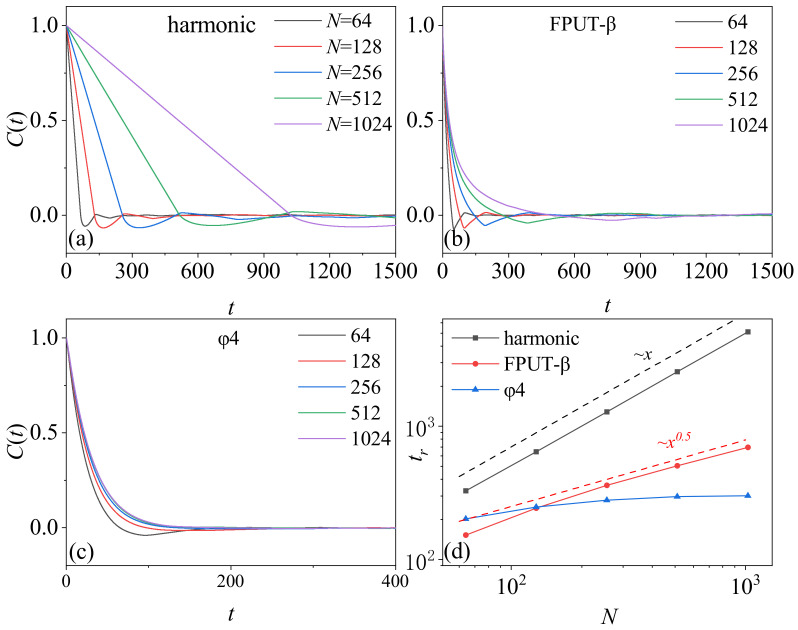
The correlation time of the global heat flow of the three models with varied particles numbers annotated in the plots. System parameters: consistent across all models. The average temperature at both ends of the heat reservoir T=0.5, and the temperature difference ΔT=0.1. (**a**–**c**) Display the correlation function, C(t), as a function of *t* for three different lattice models. (**d**) Presents the correlation time, $t_{r}$, as a function of *N* for all lattice models.

**Figure 5 entropy-26-00156-f005:**
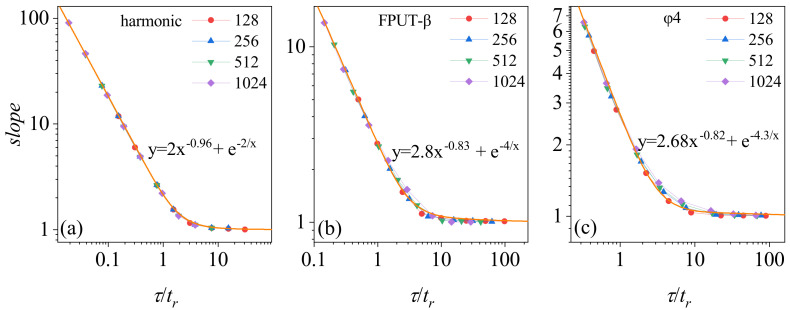
Scaling of heat flow fluctuation behavior in three model types over varied time intervals. This figure presents the scaling of heat flow fluctuation behavior in three different types of models, utilizing various time windows normalized by correlation time. System parameters: consistent across all models. The average temperature at both ends of the heat reservoir T=0.5, and the temperature difference ΔT=0.1. The number of particles are denoted in the plots. The results from molecular dynamics simulations are depicted as point plots. An orange solid line represents the fitting curve. (**a**) The harmonic oscillator model. (**b**) The FPUT-β model. (**c**) The $\phi^{4}$ model.

## Data Availability

Data are contained within the article.

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
