# Peer review of "Fluctuation Relation for the Dissipative Flux: The Role of Dynamics, Correlations and Heat Baths"

_entropy, 2024, doi:10.3390/e26020156_

Round 1
Reviewer 1 Report
Comments and Suggestions for Authors
Section 2 presents a rather technical introduction to fluctuation relations for steady states.
Notations may be familiar to experts but turn out to be difficult for non-experts. Thus,
such shortened notations hide basic notions. Just to quote a few examples which should be
improved, among others:
-the Anosov assumption should be shortly explained
-what is the meaning of SLLOD
-all symbols in Eq. (1) should be adequately explained
-the sort of Liouville equation referred to between Eqs. (2) and (3) should be explained
-a short explanation of Eq. (6) should be provided
So, it has not been easy for me to follow the detailed probabilistic arguments in Section 2. I
would recommend even a rewriting of most of (if not all) Section 2, thereby providing
improvements addressed to non-.experts. This, even if somewhat extensive, would be a
minor revision, as it would not affect the main core (devoted to new researches) of the paper.
In this connection, I suggest the authors to indicate suitable introductory reviews. Even
if they are far more qualified than me for that, let me
suggest to add (among other possible ones) the following introductory reference:
K. Mallick: “Some Recent Developments in Non-Equilibrium Statistical Physics”, Pramana
Journal of Physics, vol. 73, pages 417-451 (2009).

The most important comment:
Please, upon revising Section 2 as indicated, take care to use all necessary words and due explanations so as to avoid a too shortened section
Author Response
Thank you very much! Your very useful comments have allowed us to explain much more in detail the theory we investigate. To do that, Section 2 has been substantially revised ad extended and two appendices have been added to explain SLLOD and Anosov systems, without interrupting the flow of the main text. All symbols have been explicitly defined and examples are given to illustrate such definitions, the Liouville Eq.(2)(3) has been explained and Eq.(6), now Eq.(18) has been explicitly derived. And we have add ref.[1] to the first 5 of the original manuscript.
[1] K. Mallick: “Some Recent Developments in Non-Equilibrium Statistical Physics”, Pramana Journal of Physics, vol. 73, pages 417-451 (2009)
Reviewer 2 Report
Comments and Suggestions for Authors
This is a very interesting paper which examines decay of correlations as they relate to the steady state fluctuation theorem. It utilises 1D heat conduction models with fundamentally different behaviour in order to clarify the connection between satisfaction of the asymptotic SSFT and the system's dynamics/heat bath. Importantly, the authors demonstrate that a steady state fluctuation relation can be obtained by consideration of fluctuations as uncorrelated random variables, leading to a size-independent relation. I find this article well-presented overall, and of high quality and interest.
Nevertheless, I have a few comments/suggestions.
(1) On line 30-32, the authors claim "On the other hand, steady state fluctuation relations, apparently similar or directly derivable from transient relations, are not exact, and require proper conditions to be verified, concerning the time correlation function of the dissipation." I suggest that this claim should additionally be qualified to reflect the fact that an exact steady state fluctuation relation can be evaluated if correlations are known. For example, the authors later show an exact form in Eqn. 7 for the case of instantly decaying correlations, and it has recently been shown that a machine learning model which learns the correlations can also satisfy a non-asymptotic steady state fluctuation relation [1]. A small change such as "... are not exact unless correlations are known, or otherwise require ..." would be sufficient.
(2) In Eqn. 5, I assume $\Lambda_{0,\tau}$ is defined in a similar manner to the integrated dissipation function of Eqn. 2. The authors should clarify this.
(3) On line 135, no definition is given for $T_R$ and $T_L$. From the context, I assume these are the temperatures $T_c$ and $T_h$, respectively, shown in Fig. 1. It would be nice to be consistent in notation here with the figure.
(4) For the $\phi^4$ model as described in Eqn. 19, it is unclear to me how heat can propagate through the lattice if V(x) = 0. I note that similar models in (e.g.) refs 17 and 18 use a harmonic pair potential.
(5) Comparing to refs 23 and 24, it appears to me that there is an error in Eqn. 20 and/or the definition of $f_i$ below it. In particular, $f_i$ is defined as the force on the ith particle, but it appears to be missing a - sign in the derivative of the potential, and only includes the interaction with particle i+1 (and not i-1). Eqn. 20 then adds together $f_i$ and $f_{i+1}$, meaning that it considers particles, i, i+1 and i+2, where I would expect it to be in terms of particles i-1, i and i+1. I am not an expert on 1D lattice models though, so I may be mistaken here. I expect this does not affect the findings, but would encourage the authors to double check these equations. The variable $a$, which I take to be the lattice spacing, is also not defined here.
(6) Lines 203, 217, etc. discuss the integral of the correlation function as $t_c$, but it is plotted in Fig. 4(d) as $t_r$.
(7) On line 224-225, the authors quote "numerous studies in the equilibrium state." I suggest citing some examples here.
(8) On line 227, I suspect the reference to Fig. 4(c) should instead be to Fig. 4(d).
(9) On line 239, the RHS of the first equality appears to be missing a factor of $2\sigma^2$. This looks to be a typographical error, as the final expression is correct.
(10) With regard to the result obtained in the paragraph from lines 236 to 241, it is well known that the probability distribution is not expected to be Gaussian in general, especially near the tails of the distribution, so it is surprising that a Gaussian appears to work here. It has been shown that since the central limit theorem is only valid near the mean, FTs derived from it break down at large fields where the tails become important [2-4]. I wonder whether the agreement seen here (line 239-240) is related to the approximation in Eqn. 12 that the integrated fluxes into and out of the system are equal under a long enough time average. As highlighted by Ref. 21, this condition will not be true instantaneously, and I would expect that rare trajectories in the tail of the distribution may require longer in some cases before the condition is met, so I think this approximation may similarly break down far from equilibrium.
(11) Fig. 5 presents the first two models in the opposite order compared to other figures, and a, b, c labels are missing. It would be clearer to the reader if this were consistent, and I suspect the reference on line 263 to Fig. 5(a) is to the middle panel (harmonic model). This figure also uses $t_r$ as the correlation time where the text uses $t_c$, and $t$ where the text uses $\tau$.
(12) On line 243, by "the slope of $F(J) - J$ curve", do the authors mean the derivative of $F(J)$ with respect to $J$? Based on Fig. 3, I would expect the slope of $F(J) - J$ to converge to 0 rather than 1.
[1] S. Sanderson et al., Machine learning a time-local fluctuation theorem for nonequilibrium steady states, Progress of Theoretical and Experimental Physics, Volume 2023, Issue 8, August 2023, 083A01. https://doi.org/10.1093/ptep/ptad102
[2] D. J. Searles
D. J. Evans; The fluctuation theorem and Green–Kubo relations. J. Chem. Phys. 8 June 2000; 112 (22): 9727–9735. https://doi.org/10.1063/1.481610[3] E. M. Sevick et al., Fluctuation Theorems, Annual Review of Physical Chemistry, 2008 59:1, 603-633. https://doi.org/10.1146/annurev.physchem.58.032806.104555 (see Section 3.4)
[4] D. J. Evans et al., Application of the Gallavotti-Cohen fluctuation relation to thermostated steady states near equilibrium, Phys. Rev. E, 71, 056120. https://doi.org/10.1103/PhysRevE.71.056120 (see Section V)
Quality of English language in this article generally good. I noticed some minor typographical errors throughout, but overall the meaning is clear.
I also note that the 4s in $\phi^4$ appear to normally be written as a superscript in the articles referenced.
Author Response
Thank you very much for your positive comments on our paper. Many thanks for your careful and detailed report. All you points are correct, we have revised them following your comments. We are very sorry for made so many typo errors. We've corrected the sign of f_i. We've uniformed the quantity t_c as t_r. We've cited several references for your point 7.
For your point 2, we have rewritten the Section 2, as also for response other referees.
For your point 10, it is indeed a profound question, yet we must admit a lack of complete confidence in providing a clear answer to your concerns. Firstly, it's important to note that the Gaussian distribution we observed is a direct result of our numerical simulations, as illustrated in Figure 2. This is similar to previous studies to this problem such as [1], which also indicated that heat flow fluctuations are close to a Gaussian distribution, albeit with a much larger temperature difference in the heat baths compared to our study, thus making our results appear more Gaussian. Theoretically, since we are studying the steady-state fluctuation theorem – more specifically, the dissipative steady-state relation of heat flow – the overall heat flow is the sum of local heat flows across individual lattices. The temporal accumulation of heat flow, being a sum of fluctuated variables. These facts lend to the heat-flow fluctuations more readily to conforming with the Central Limit Theorem. Indeed, the Gaussian distribution is a special type of large deviation function. When one conducts a second-order expansion on the rate function of the large deviation function, it can degenerate back into a Gaussian distribution (refer to [2, 3]). This implies that, aside from the exponential distribution, the Gaussian distribution can also satisfy the fluctuation theorem. It is likely that the Gaussian distribution represents a characteristic distribution under the steady-state fluctuation theorem. We concur that this may be related to the approximation in our Equation 12 (Equation 26 in the revised paper), where it's assumed that the integrated flows into and out of the system are equal under a sufficiently long time average. However, unfortunately, we have not explicitly derived that the overall heat flow fluctuation must be Gaussian. We hope to leave this question open for further research in the future.
[1] Lepri, S. et al Physica D: Nonlinear Phenomena 1998, 119, 140–147
[2] Luca Peliti and Simone Pigolotti, "Stochastic Thermodynamics: an Introduction", 2021, p. 132
[3] Sevick E.M et al., "Fluctuation Theorems", Annual Review of Physical Chemistry, Vol. 59, 2008, p. 19
We have cited these references in the revised paper.
Reviewer 3 Report
Comments and Suggestions for Authors
The authors present a nice article on fluctuation relations for dissipation functions and derive a form of steady state fluctuation relation from the well known transient forms.
While the physical intuition makes sense, section 2 needs a lot of revision to be clearly communicated to the reader. Specific suggestions are:
(1) the authors define time reversal invariance as a transformation whose double application leads to identity. This definition is ambiguous and must be clarified. Double application must consist of one back mapping for this definition to work.
(2) In (6), the departure parameter $\eps$ is not explained and as such introduced ad hoc.
(3) Eq. (7) essentially represents a mean over a sub-ensemble of trajectories. How does one understand the distribution properties for this sub-ensemble? Specifically are higher order moments guaranteed to follow the same original distribution?
I would be happy to recommend this for publication after the authors convincingly address the questions and revise their section 2 accordingly.
Author Response
Our response:
Many thanks. Your useful comments have allowed us to explain much more in detail the theory we investigate.
For point (1), an appendix has been added.
For point (2) and (3), section 2 has been substantially revised, and it is now clear what the various terms mean, and the various equations are more clearly derived.
Round 2
Reviewer 2 Report
Comments and Suggestions for Authors
The authors have sufficiently addressed my comments.
On point 10, I agree that this would be an interesting avenue of future research. I also agree that while it may not be the case for general systems, a distribution which is close to Gaussian makes sense in the context of heat flow, especially near equilibrium.
Comments on the Quality of English LanguageSome minor typographical/grammatical errors are still present, but the meaning is clear. I assume these will be picked up in the final editorial process.
Reviewer 3 Report
Comments and Suggestions for Authors
Questions and concerns are addressed. Accept in present form.